# Digital Competence of Higher Education Teachers at a Distance Learning University in Portugal

**José António Moreira** [1,*], **Catarina S. Nunes** [2] **and Diogo Casanova** [3]

1   CEG, Department of Education, Universidade Aberta, 1269-001 Lisbon, Portugal
2   Department of Sciences and Technology, Universidade Aberta, 1269-001 Lisbon, Portugal;
    catarinas.nunes@uab.pt
3   LE@D, Department of Education, Universidade Aberta, 1269-001 Lisbon, Portugal; diogo.casanova@uab.pt
*   Correspondence: jmoreira@uab.pt

**Abstract:** The Digital Education Action Plan (2021–2027) launched by the European Commission aims to revolutionize education systems, prioritizing the development of a robust digital education ecosystem and the enhancement of teachers' digital transformation skills. This study focuses on Universidade Aberta, Portugal, to identify the strengths and weaknesses of teachers' digital skills within the Digital Competence Framework for Educators (DigCompEdu). Using a quantitative approach, the research utilized the DigCompEdu CheckIn self-assessment questionnaire, validated for the Portuguese population, to evaluate teachers' perceptions of their digital competences. A total of 118 teachers participated in the assessment. Findings revealed that the teachers exhibited a notably high overall level of digital competence, positioned at the intersection of B2 (Expert) and C1 (Leader) on the DigCompEdu scale. However, specific areas for improvement were identified, particularly in Digital Technologies Resources and Assessment, the core pedagogical components of DigCompEdu, which displayed comparatively lower proficiency levels. To ensure continuous progress and alignment with the Digital Education Action Plan's strategic priorities, targeted teacher training initiatives should focus on enhancing competences related to Digital Technologies Resources and Assessment.

**Keywords:** digital competence; higher education; DigCompEdu; distance education

## 1. Introduction

Involved in processes of change, often justified by diffuse policies and isolated instrumental measures, higher education institutions (HEIs) in Portugal are marked by a traditional and elitist educational culture and have been confronted with the need to innovate, especially in terms of pedagogical practices.

Conversely, innovating from a pedagogical point of view in higher education (HE) implies reconfiguring the pedagogical cultures with the incorporation of new spaces and virtual environments in HE.

Several didactic proposals can support innovation processes, for example, approaches based on research, collaboration, project development, problem solving or community intervention, as is suggested in the Open University's recent publication Innovating Pedagogy 2023 [1].

In this context, the advancement of digital technologies has assumed an important role, which has recently been amplified by the COVID-19 pandemic. However, whilst the use of these technologies has been envisaged to transform teaching and learning practices, what often happens is that their introduction allows for the transfer and reproduction of existing practices rather than truly transforming them.

From a transformative perspective of HE, pedagogy is a place of knowledge production (not mere reproduction) for both students and teachers [2]. A transformative pedagogy

implies methodologies that are open to reflection on learning content and processes, negotiation of meanings and decisions, and the construction of broad visions of knowledge, which expands and complexifies the role of the teacher, as the teacher becomes a "designer" and an "architect" [3,4] not only of learning scenarios in physical spaces, but also in online virtual spaces.

However, for these methodologies to be effective, teachers and students need to adapt to the new spaces and times of education and learn to incorporate the digital and the virtual into their practices. The integration of the digital space cannot only imply the aforementioned reproduction of conservative practices for emerging virtual environments; on the contrary, this integration must envisage the integration of innovative methodologies that develop in ubiquitous, natural, constructed or virtual learning scenarios through mobile devices, connected to wireless communication networks, sensors and geo-localization mechanisms, allowing the formation of virtual networks between people, objects and situations. Teachers thus face an added challenge: they must be able to incorporate the digital into their practices in a critical, reflexive and pedagogically intentional way. Recently, the European Commission launched an initiative, the Digital Education Action Plan (2021–2027) [5], to address the challenges facing European education systems, which defines two strategic priorities: (a) promoting the development of a highly effective digital education ecosystem; and (b) strengthening digital competences and skills for digital transformation.

Assuming, therefore, that the reinforcement of the digital competences of higher education teachers is a political and social priority in the European space, which has gained even greater relevance during the last year, we developed the present study, which aims to evaluate the level of digital competence of the teachers of the Universidade Aberta (UAb) of Portugal, identifying the areas of competence with greater weaknesses and, based on this diagnosis, pointing out possible training responses according to the level achieved. This has been a policy of UAb, the Portuguese open and distance learning university, which has a history of training higher education teachers directly or indirectly on this issue of teacher digital competences. This tradition, an identity mark of the institution since the beginning of the 21st century, has been emphasized since 2007, with the transition of teaching and learning processes to an online digital environment and the creation of its virtual pedagogical model (VPM). From then on, all UAb's teaching staff participated regularly, not only in training courses on the VPM but also in courses related to the creation of online teaching and learning scenarios and processes and, over the last decade, several other training opportunities have been offered to teaching staff as new academic staff joined the university. More recently, in this post-pandemic period, a set of new online training courses has been offered in a formal environment, with the attribution of micro-credentials, and informally, through workshops.

It is therefore in this context of investment in pedagogical-digital training that UAb, through its Rectorate, decided to carry out a study to evaluate the digital competences of its teachers, and, based on these results, to verify the training needs in each of its departments.

This evaluation was based on a questionnaire developed by the EU Science Hub (Science and Knowledge Service of the European Commission), whose main component is based on a self-reflection tool—DigCompEdu CheckIn—developed based on the European Digital Competence Framework for Educators (DigCompEdu), which allowed a response to the requested purposes, specifically, to identify: (i) proficiency levels and overall average scores of teachers, (ii) proficiency levels by areas of digital competence, and (iii) proficiency levels by age group, gender, teaching time, time of use of digital technologies and virtual environments and organization unit.

## 2. DigCompEdu and DigCompEdu CheckIn

DigCompEdu, the framework used in this study, was developed by a team of education experts and practitioners led by the Joint Research Centre (JRC B.4) in Seville and was translated into Portuguese in 2018 by Lucas and Moreira [6]. The framework provides a

common language and reference on what it means to be digitally competent, offering a set of useful descriptors for teachers' (self-)assessment and professional development.

The Digital Competence of Educators (DigCompEdu) model [7] consists of six areas, summarized in Figure 1.

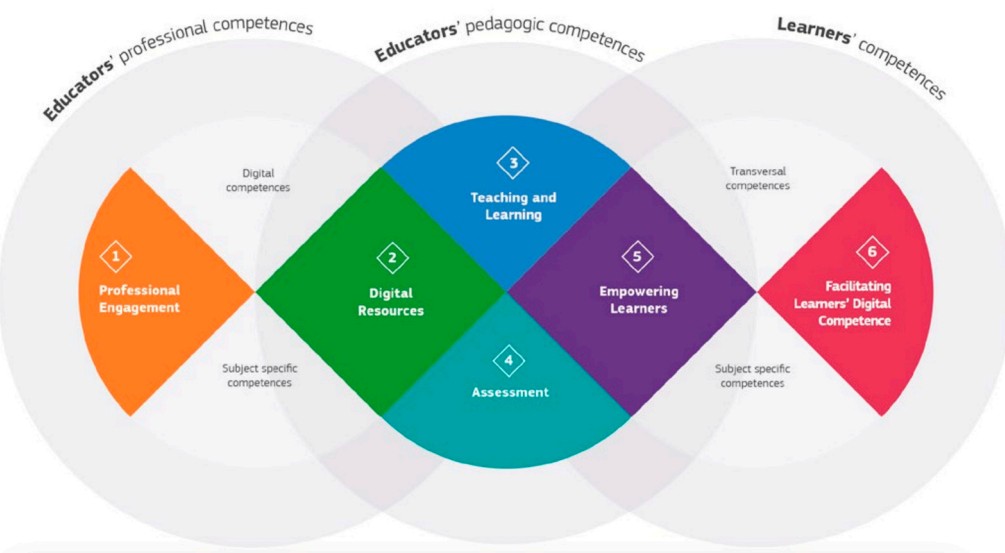

**Figure 1.** Model DigCompEdu source: Redecker [7].

These competences are organized into three dimensions, each addressing essential aspects of teachers' digital proficiency. The first dimension, "Professional Engagement", encompasses Competence Area 1. It revolves around teachers' adept use of digital technologies in their professional setting. They employ these tools to interact and collaborate effectively with colleagues, students, and parents or guardians, as well as to support their ongoing professional development.

Moving on to the second dimension, it focuses on the pedagogical aspects specifically relevant to the teaching and learning process. Within this dimension, we find Competence Areas 2 to 5: "Digital Technologies and Resources", "Teaching and Learning", "Assessment", and "Empowering Learners." Competence Areas 2 to 4 delve into the digital competences teachers require to skillfully select, create, and adapt digital resources. Additionally, they encompass the ability to design, implement, and assess effective teaching and learning experiences using these resources. Competence Area 5, however, centers on the digital competences necessary to place learners at the core of these processes, fostering student empowerment in a digitally enriched educational environment.

The third dimension centers around Competence Area 6, "Facilitating Learners' Digital Competence." This dimension reflects the critical skills needed to empower learners with the digital fluency necessary for autonomous, critical, and creative use of digital technologies.

In total, these dimensions encompass twenty-two competences, showcasing the comprehensive skill set required of teachers in their digital journey. The competences are organized into six progressive levels of ICT appropriation, ranging from beginner to innovative levels (Figure 2). As educators advance through these levels, they gain a deeper understanding and mastery of digital tools, ultimately transforming their teaching approaches and positively impacting their students' learning experiences.

In this context, the initial two levels, "Newcomer" (A1) and "Explorer" (A2), involve teachers assimilating new information and developing basic digital practices. As they progress to the intermediate levels, "Integrator" (B1) and "Expert" (B2), they begin to apply the knowledge they have gained, actively seeking to expand and refine their digital teaching practices. Finally, at the most advanced levels, "Leader" (C1) and "Pioneer" (C2), teachers demonstrate their proficiency by not only applying their knowledge but

also sharing it with others, critically analyzing existing practices, and actively developing innovative approaches.

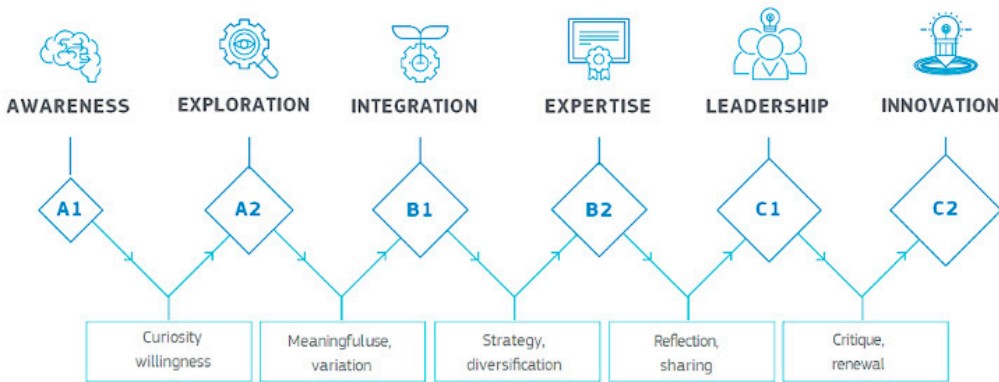

**Figure 2.** DigCompEdu Proficiency Levels source: Redecker [7].

To facilitate this journey of self-assessment and professional growth, the DigCompEdu Check-In was devised as a self-reflection tool. Developed collaboratively by JRC B4 in partnership with researchers and educators from various countries, this tool comprises one statement for each of the 22 competences proposed by DigCompEdu.

The DigCompEdu Check-In adopts a scoring system where each answer provided by the teacher is rated on a scale from 0 to 4. This approach allows for a comprehensive evaluation of the teacher's digital proficiency. Notably, a personalized feedback report is generated for each answer, providing valuable insights and practical tips for the teacher to enhance their proficiency level in specific areas.

The primary aim of the DigCompEdu Check-In is to encourage teachers, whether individually or in groups, to engage in self-reflection on their digital competence. By leveraging the detailed feedback report, educators can identify areas for improvement and design tailored paths of professional development. This iterative process of improvement fosters the ongoing growth and refinement of digital teaching practices.

The questionnaire authors differentiate respondents based on their choices, particularly those who predominantly select the first option and are classified as "newcomers." On the other end of the spectrum are the "pioneers", who must consistently opt for the highest option in at least two-thirds of the 22 competences to achieve the highest proficiency level.

It is worth noting that the DigCompEdu Check-In, along with its allocated score ranges for different competence levels, has undergone validation through various European studies. In Portugal, Dias-Trindade et al. [8] have thoroughly validated the psychometric properties of the tool, affirming its reliability and effectiveness. For the present study, this version of the tool, with slight modifications related to construct validity analyses (exploratory and confirmatory factor analysis), has been utilized.

The rigorous validation process ensures that the DigCompEdu Check-In provides accurate and dependable assessments, empowering teachers to navigate their digital competence development effectively and, ultimately, enhance their instructional practices in the digital age.

## 3. Materials and Methods

The empirical component of the research proposed follows a quantitatively oriented procedure by placing emphasis on the teachers' perception of issues related to their digital teaching competences.

As already noted, the main objective of this study is to evaluate the level of digital proficiency of teachers at the UAb and to identify their training needs in the six dimensions considered. And this is an increasingly important issue, because, both nationally and internationally, the growing awareness that teachers must keep up with digital evolution and train themselves in the skills to use digital technologies in different educational

environments, physical and virtual, has been a reality. Nonetheless, we often find the perception, on the part of many teachers, that, on the one hand, the existing specialized training does not accompany their real needs and, on the other hand, the many activities, especially scientific ones, in which they are involved interfere with the time they have available to dedicate to training in the pedagogical-digital area. It should be noted that these perceptions result, for the most part, from superficial opinions and assessments and not so much from scientific studies that verify how most of the teaching staff is in terms of digital skills.

And it is in this context that work such as that being developed at the EU Science Hub, a department of the European Union, which has sought to identify the needs of teachers through the development of models, questionnaires and reports that support the work developed in this area, becomes relevant, as does that of the European Digital Education Hub.

### 3.1. Questionnaire

The questionnaire applied consists of two sections. The first section collects the teachers' self-reflection on their digital competence; the second section collects sociodemographic data, including gender, age, organic unit, and years of using digital technologies in the teaching and learning process.

As already mentioned in the previous section, the questionnaire used to assess digital competences was the DigCompEdu Check-In, in the version validated and translated for the Portuguese population by Dias-Trindade et al. [8], which presents slight differences from the original, namely, in some of the questions to adjust to the context of online teaching and in the number of competences under analysis, and analyzes 21 competences, instead of the original's 22. The internal consistency of the questionnaire was evaluated using Cronbah's Alpha, and the result obtained ($\alpha = 0.939$) indicates an excellent internal consistency.

For each of the 21 competences, a statement (item) is presented, and participants must select one of the options that best characterizes their position towards that same statement.

For each of the items, the same point levels are assigned, ranging from 0 for the first hypothesis to 4 points for the last. In this sense, the total score of the test is 84 points, dividing the digital competence levels in six (Table 1).

**Table 1.** Digital competence levels of the DigCompEdu CheckIn Questionnaire.

| Digital Competency Level | Score |
| --- | --- |
| Al—Newcomer | Below 19 points |
| A2—Explorer | From 19 to 32 points |
| B1—Integrator | From 33 to 47 points |
| B2—Specialist | 48 to 62 points |
| C1—Leader | From 63 to 77 points |
| C2—Pioneer | Above 77 points |

A detailed individual feedback report is provided at the end of the questionnaire. One of the main purposes of the questionnaire is to allow the teacher to reflect on his/her digital competence and, based on this report, to plan professional development pathways for continuous improvement.

### 3.2. Procedure

The questionnaire, hosted originally on the European Commission's online platform (https://ec.europa.eu/eusurvey/runner/Pesquisa_UAb_PT, accessed on 13 August 2023), was made available to teachers during the year 2022. Teachers' participation was consensual and voluntary.

### 3.3. Sample

Teachers from the four departments of the UAb and the Unit for Lifelong Learning were invited to complete the questionnaire. Of the universe of 140 teachers, 107 completed the questionnaire (76.4%), with the majority being female (54.2%).

Of these, the median age group is between 50 and 59 years, with just under 50% falling within this age range. In fact, the highest percentage of teachers is in the 50–59 age group, followed by the 40–49 and 60+ age groups. These three groups together account for 95.3% of the sample. Only 4.7% of the sample is made up of teachers aged 30 to 39.

Regarding the distribution of the sample by organic unit in Table 2, it is observed that a greater number of responses were obtained from the Department of Science and Technology (36 of 39 teachers). Despite being fewer in number, it was observed that all teachers from the Department of Education and Distance Learning responded to the questionnaire. The Department of Social Sciences and Management, with the largest number of teachers in the institution, also showed a lower participation (23 of 49 teachers).

**Table 2.** Distribution of the sample by organic unit.

| Organic Unit | Frequency | Percentage |
| --- | --- | --- |
| Department of Science and Technology (DCeT) | 36 | 33.6 |
| Department of Social and Management Sciences (DCSG) | 23 | 21.5 |
| Department of Education and Distance Learning (DEED) | 25 | 23.4 |
| Department of Humanities (DH) | 22 | 20.6 |
| Lifelong Learning Unit (UALV) | 1 | 0.9 |
| TOTAL | 107 | 100.0 |

Concerning the years of teaching, most teachers had been teaching for more than 20 years (55.1%), and it was not possible to show whether they taught only at the UAb and in the distance education modality or if they had previous teaching experiences. Teachers were mostly experienced, with 72.9% of the sample having more than 15 years of service.

On the question of how many years they had been using digital technologies and virtual environments in the teaching and learning process, it appears that more than 50% of teachers had been using digital technologies and virtual environments for more than 14 years, while only 10% had been doing so for less than six years. These answers were expected because we were analyzing the results of the teachers of an online higher education institution, and they reveal the experience of the university's teachers regarding the integration of digital in their pedagogical practices.

### 3.4. Data Analysis

SPSS statistical software (IBM SPSS® version 25) was used for data analysis. Descriptive analyses were based on absolute and relative frequencies and inferential analyses on the non-parametric Kruskal–Wallis test (followed by Pairwise tests with Bonferroni correction), considering a significance level of 0.05.

## 4. Results

The results of this study offer an insightful and comprehensive portrait of teachers' digital technology usage. We synthesized the data to provide a comprehensive overview of the teachers' proficiency levels and mean scores across various digital competence areas. Additionally, we analyzed the data to identify potential variations based on several demographic factors, including age group, gender, duration of teaching service, duration of digital use, and the specific organic unit to which they belong.

### 4.1. Overall Proficiency Levels

Figure 3 shows the distribution of teachers across the different proficiency levels.

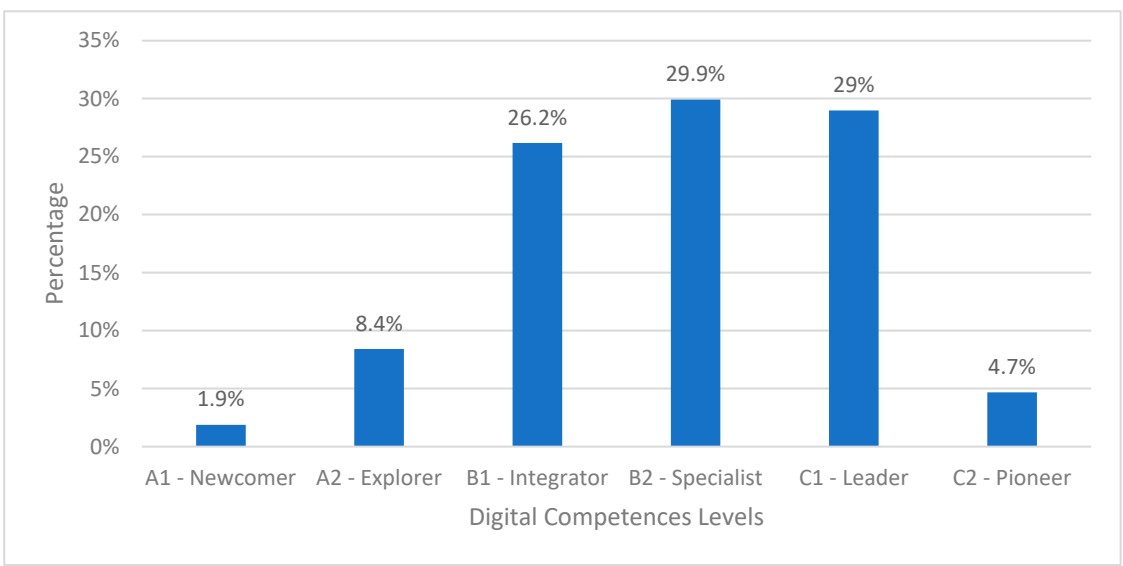

**Figure 3.** Distribution of digital competence levels (general diagnosis).

The visual analysis of the data illustrates significant fluctuations in responses, indicating an uneven distribution of teachers across different proficiency levels in their use of digital technology. Notably, we observe a homogeneous distribution at levels B1, B2 and C1, while there are notably lower distributions at the extreme proficiency levels, A1, A2 and C2.

It is noteworthy that the cumulative proportion of responses corresponding to the two lowest levels of digital proficiency accounts for only 10.3% of the total sample. In contrast, the accumulation of proportions for levels A, B and C reveals that level B (B1 + B2) constitutes a substantial 56.1% of the responses. Level A (A1 + A2) encompasses 10.3% of the responses, while C (C1 + C2) represents 33.7%. These findings hold significant implications for the overall level of digital proficiency among teachers.

*4.2. Proficiency Levels by Areas of Competence*

In Area 1-Professional Engagement (Figure 4), we observed that 11.2% of teachers exhibit basic digital proficiency (levels A1 and A2), relying on digital technologies for fundamental communication, interaction, and collaboration with colleagues and students. The largest segment of the sample, comprising 68.2% of teachers, demonstrated intermediate proficiency levels. These individuals effectively and responsibly employ digital tools to enhance communication within the institution and support their professional development.

Additionally, we identified a group of highly proficient teachers (20.6%) who are advance in utilizing digital technologies. They actively engage in reflective practices to enhance institutional communication and consistently leverage digital tools to support their ongoing professional growth.

Upon analyzing responses related to this dimension, we discovered that a significant portion of teachers (52.3%) skillfully integrate various digital solutions to communicate more effectively based on their specific objectives. Furthermore, a majority of educators had been actively enhancing their digital skills either through self-directed learning or through collaborative discussions with colleagues, focusing on innovating and improving educational practices.

Regarding online training involvement, 44.5% of respondents reported participating in training sessions multiple times, while 42.1% engaged in such training very frequently. This highlights the teachers' proactive approach to continuous learning and professional development in the digital realm.

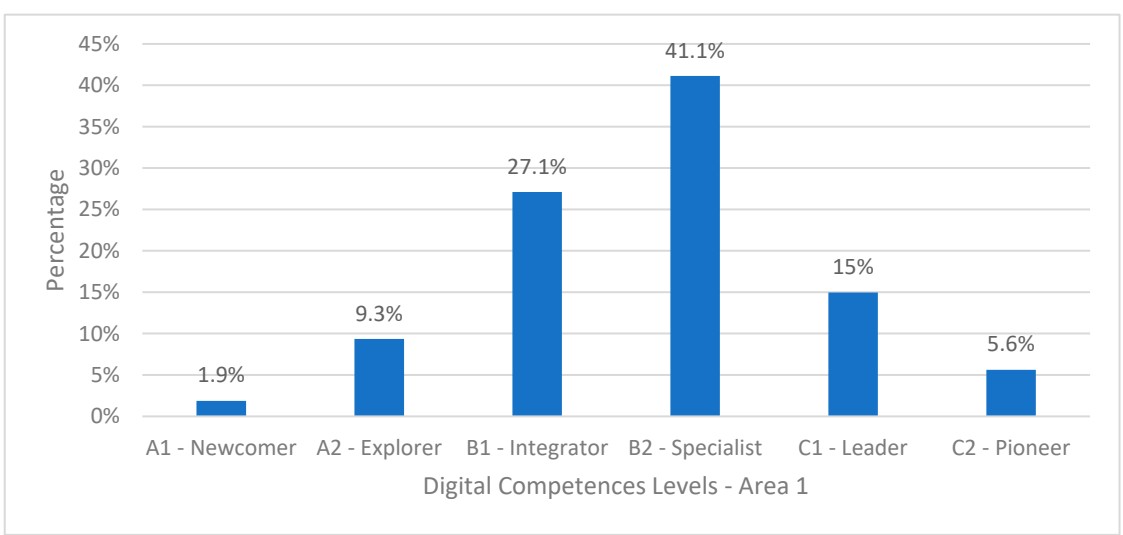

**Figure 4.** Distribution of competence levels for Area 1—Professional Engagement.

An encouraging finding was that approximately 90% of teachers demonstrated critical thinking in their utilization of digital resources. This indicates a high level of discernment and thoughtfulness in selecting and employing digital tools, ensuring their effective and meaningful integration in educational settings.

In Area 2-Digital Technologies Resources (Figure 5), we observed a predominant concentration of teachers at intermediate levels, accounting for 56.1% of the sample. Notably, a significant portion of these teachers excelled at level B1, demonstrating their research proficiency in identifying and evaluating digital resources that they can modify and adapt to suit their instructional needs. Furthermore, they play an essential role in selecting digital resources, recommending their use to students and critically assessing the reliability and suitability of these resources for their institution's pedagogical project.

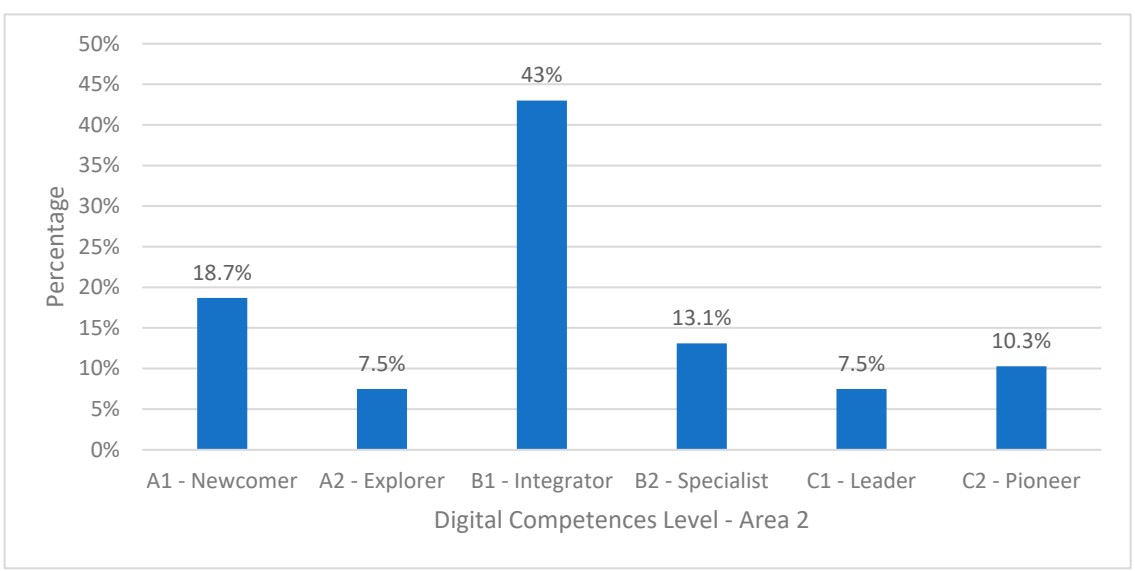

**Figure 5.** Distribution of competence levels for Area 2—Digital Technologies Resources.

Among the respondents, 26.2% displayed proficiency at the initial levels. These teachers primarily rely on simple internet search strategies to locate digital content relevant to the teaching and learning process. However, they do not actively engage in modifying or sharing these resources.

Interestingly, only a minority of teachers, comprising 17.8%, demonstrated advanced proficiency levels in this area. These educators are highly adept at evaluating, creating and publishing interactive digital content to enhance the teaching-learning process.

Upon analyzing the responses, we noted that teachers displayed significant confidence in using digital technologies and resources. Around 90% of respondents reported collaborating through shared portfolios and collaborative environments, fostering a sense of community and knowledge exchange. Approximately 41% of these educators actively utilize networks and sharing platforms to exchange ideas and materials with their peers, promoting a culture of innovation and knowledge dissemination.

However, one concerning aspect of our analysis relates to the use of security mechanisms to protect sensitive content. Only 33% of teachers reported using conscious measures to safeguard documents and files when sharing them. This indicates a potential area for improvement in ensuring data security and privacy.

In Area 3—Teaching and Learning (Figure 6), 89.6% of teachers are at intermediate (55.1%) and advanced (34%) levels, showing that they have no difficulties in using digital technologies in teaching and learning processes, either in terms of promoting student interaction and monitoring or in promoting collaborative learning strategies.

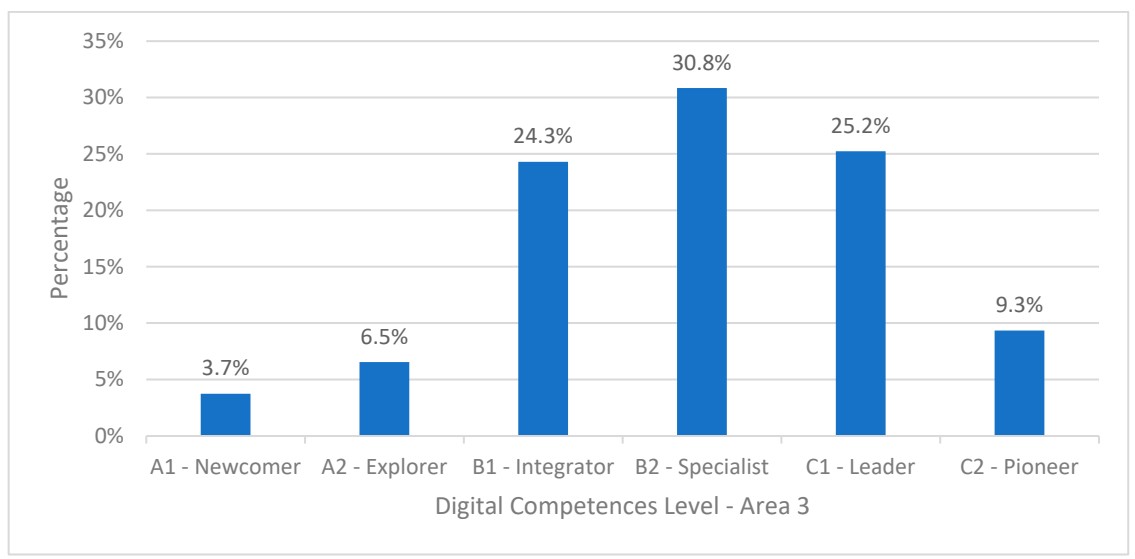

**Figure 6.** Distribution of competence levels for Area 3—Teaching and Learning.

On the other hand, 10.2% of teachers are still at the initial levels, as they have some difficulties in using digital technologies to promote pedagogical practices based on interaction and collaborative learning.

A closer reading of the answers to the questions in this dimension allows us to realize that most teachers use digital technologies to systematically improve the teaching and learning process, and about 25% reveal that they seek to implement innovative strategies. As an online institution, it is natural for teachers to monitor student activities in the different collaborative environments of the networked digital ecosystem. It should also be noted that most of the teachers surveyed encourage students to use digital technologies to carry out group work. In turn, regarding the implementation of active methodologies in their practices, it appears that about 90% use digital technologies to develop these active methodologies.

Finally, approximately 80% of teachers actively promote learning activities involving students' creation of digital content, like videos, audio, photos, digital presentations, blogs or wikis, within their curricular units. This reflects the recurring use of digital technologies by teachers and the ample opportunities provided to students for content creation using these tools. Hence, digital technologies are not just mediums for information dissemination but potent pedagogical resources that enrich and enhance online educational practices.

Contrary to the results of most of the studies carried out in this field of teacher digital competences, Area 4—Assessment (Figure 7), shows much higher levels of proficiency than other research [9,10].

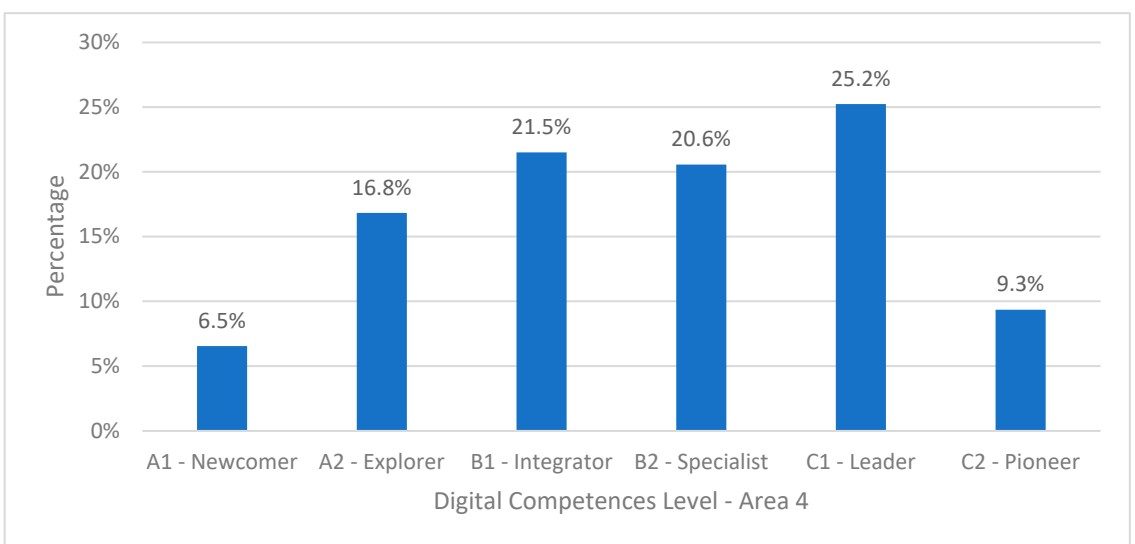

**Figure 7.** Distribution of competence levels for Area 4— Assessment.

In fact, and despite presenting lower values than the other areas under analysis, 76.6% of the teachers in the sample are at an intermediate (42.1%) and advanced (34.5%) level of digital proficiency, with teachers revealing that they frequently and effectively use digital tools to plan, implement and evaluate educational processes.

On the other hand, 23.4% of the teachers at the initial levels (A1 and A2) make incipient use of technologies in assessment strategies.

From the analysis of the responses in this dimension, it stands out that more than 80% of teachers use different software and digital technologies to check student progress and provide more efficient feedback.

Considering that this analysis is carried out with teachers who work exclusively in virtual environments and who use digital assessment software, it is not surprising that these values are higher than the values present in most of the studies available in this area; however, it would be expected that the percentage of teachers positioned at level A would be even lower, since digital assessment is an intrinsic and indispensable component of the institution's virtual pedagogical model. Finally, regarding the promotion of learning activities that imply the creation of digital content by students, such as videos, audio, photos, digital presentations, blogs or wikis, about 80% of the teachers reported they promote this type of activity in their curricular units. It can be seen, therefore, that teachers use digital technologies regularly and, in most curricular units, students have the possibility to create new content using these technologies. In essence, digital technologies are not simply employed as a medium for information dissemination, but are harnessed as powerful pedagogical resources that enrich and enhance online educational practices.

In area 5—Empowering Learners (Figure 8), 77.5% of teachers position themselves at intermediate (43%) and advanced (34.5%) levels, with practices that favor accessibility and inclusion to promote active and collaborative learning methodologies that place students at the center of these practices. Only 21.5% of teachers are at the initial levels, and although they also seek to promote the strengths present in the virtual pedagogical model, interaction and inclusion, they have more difficulty in developing these practices that place the student at the center of the teaching and learning process.

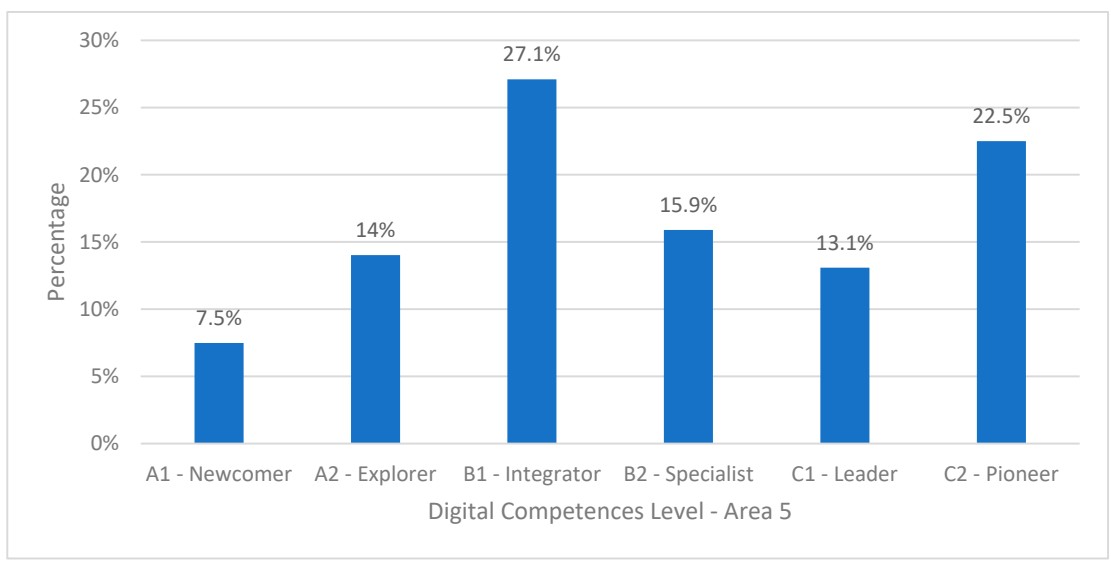

**Figure 8.** Distribution of competence levels for Area 5—Empowering Learners.

Reading the answers in this dimension, we also realize that most teachers analyze the available information regularly to identify students who need additional support. Of these, more than 75% also do so during the teaching and learning process, while 20% only analyze relevant academic information, for example, on performance and grades. Most lecturers, about 90%, also tried to solve the problems identified by students related to the activities in digital format. The answers to the questions in this dimension point to an active and monitoring participation of the teacher during the teaching and learning process.

Area 6-Facilitating Learners' Digital Competence (Figure 9), is the one in which we find lower values of digital proficiency, with 38.3% of teachers positioned at the initial levels, mainly at the newcomer level (A1), that is, they develop few strategies to promote students' digital competences. Teachers who are at intermediate levels represent 46.7% of the sample, and they promote strategies for the development of these skills, encouraging content creation and digital problem solving. Finally, 15% of teachers are at advanced levels, being able to promote students' digital competences critically and innovatively, strengthening their autonomy and security in the use of digital technologies.

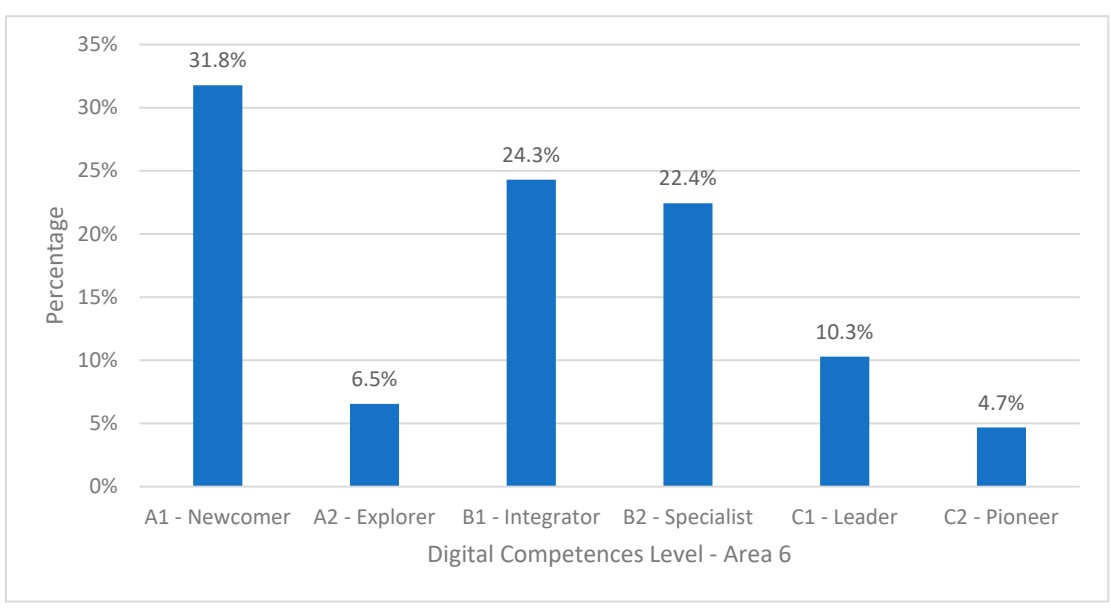

**Figure 9.** Distribution of competence levels for Area 6—Facilitating Learners' Digital Competence.

By reading the answers in this dimension, we also realize that most teachers, about 60%, discuss the quality of information, seeking to help students distinguish between possible reliable and unreliable sources. It is also interesting to realize that few teachers work on the issue of online safety, only 23%, because most of them say it is not their responsibility. It is noteworthy that only 20% of teachers encourage students to use digital technologies creatively to solve concrete problems, stating that the opportunity to do so does not always arise. Here, one can question whether the use of the traditional paper-based assessments or exam at the UAb is hampering the more creative use of digital software to realize the assessment process.

Continuing the statistical analyses, the results of the general diagnosis are presented below, cross-referencing them with the variables describing the teachers' profiles. In order to identify significant differences between the dimensions of the digital competence areas (A1 to A6), the Kruskal–Wallis test (followed by Pairwise tests with Bonferroni correction) was used to compare the value of the scores, and $p < 0.05$ was considered statistically significant. The same approach was taken for the general diagnosis score. Variables related to the profile of the respondents were compared: age group, gender, time of teaching, time of use of technologies and virtual environments and organic unit.

In the general diagnosis of digital competences (Table 3), stratified by age group, it is interesting to note that the highest concentration of teachers is found at levels B2 (Expert) and C1 (Leader), with emphasis on the 40 to 49 age group, with a value of 43.8% at level B2, and for the 50 to 59 age group a value of 36.7%. In addition, the youngest teachers (30 to 39) have lower levels of proficiency, with a higher incidence at level A2 (Explorer) and the oldest teachers (60 years or older) have a value of 14.3% at level C2 (Pioneer), higher than the other bands at this level.

**Table 3.** General diagnostic digital skill levels by age group.

| | **DIGITAL SKILLS LEVEL (GENERAL DIAGNOSIS)** | | | | | | **TOTAL** | |
|---|---|---|---|---|---|---|---|---|
| **Age Group** | **A1: Newcomer** | **A2: Explorer** | **B1: Integrator** | **B2: Specialist** | **C1: Leader** | **C2: Pioneer** | **N** | **%** |
| 30 to 39 years | 20.0% | 40.0% | -- | 20.0% | 20.0% | -- | 5 | 4.70% |
| 40 to 49 years | -- | 3.1% | 25.0% | 43.8% | 25.0% | 3.1% | 32 | 29.9% |
| 50 to 59 years | 2.0% | 6.1% | 26.5% | 26.5% | 36.7% | 2.0% | 49 | 45.8% |
| 60 and above | -- | 14.3% | 33.3% | 19.0% | 19.0% | 14.3% | 21 | 19.6% |
| **TOTAL** | 1.9% | 8.4% | 26.2% | 29.9% | 29.0% | 4.7% | 107 | 100.0% |

In the general diagnosis of digital competences (Table 4), stratified by gender, the results are very similar; however, the male gender presents slightly higher values concerning the two levels of higher digital proficiency (C1 and C2), 38.8% and 29.3%, respectively. The Kruskal–Wallis test also shows that the distribution of the score for Area 2 (Digital Technologies and Resources) reveals statistically significant differences between genders, with a higher score for males ($p = 0.003$).

**Table 4.** General diagnostic digital skill levels by gender.

| | **DIGITAL SKILLS LEVEL (GENERAL DIAGNOSIS)** | | | | | | **TOTAL** | |
|---|---|---|---|---|---|---|---|---|
| **Gender** | **A1: Newcomer** | **A2: Explorer** | **B1: Integrator** | **B2: Specialist** | **C1: Leader** | **C2: Pioneer** | **N** | **%** |
| Female | -- | 8.6% | 29.3% | 32.8% | 25.9% | 3.4% | 58 | 54.2% |
| Male | 4.1% | 8.2% | 22.4% | 26.5% | 32.7% | 6.1% | 49 | 45.8% |
| **TOTAL** | 1.9% | 8.4% | 26.2% | 29.9% | 29.0% | 4.7% | 107 | 100.0% |

The description of the levels of digital competences (Table 5), taking as a parameter the time teaching, also shows that teachers with more time in service (more than 36 years) are the ones with the highest percentage values, being at the advanced level C1 with a value of

66.7%, and at the intermediate level B2 the teachers with 5 to 10 years of service (66.7%) and 11 to 15 years of service (50%) have the highest values. Once again, these results are "out of line" with other studies.

**Table 5.** General diagnostic digital skill levels by teaching time.

| Time in Teaching | DIGITAL SKILLS LEVEL (GENERAL DIAGNOSIS) | | | | | | TOTAL | |
| | A1: Newcomer | A2: Explorer | B1: Integrator | B2: Specialist | C1: Leader | C2: Pioneer | N | % |
|---|---|---|---|---|---|---|---|---|
| Less than 5 years | 11.1% | 33.3% | 33.3% | 11.1% | 11.1% | -- | 9 | 8.4% |
| 5 to 10 years | -- | -- | -- | 66.7% | 33.3% | -- | 6 | 5.6% |
| 11 to 15 years | -- | 7.1% | 28.6% | 50.0% | 14.3% | -- | 14 | 13.1% |
| 16 to 20 years | -- | 5.3% | 26.3% | 31.6% | 36.8% | -- | 19 | 17.8% |
| 21 to 25 years | 4.2% | -- | 29.2% | 25.0% | 37.5% | 4.2% | 24 | 22.4% |
| 26 to 30 years | -- | 8.3% | 33.3% | 25.0% | 33.3% | -- | 12 | 11.2% |
| 31 to 35 years | -- | 15.0% | 20.0% | 25.0% | 20.0% | 20.0% | 20 | 18.7% |
| Over 36 years | -- | -- | 33.3% | -- | 66.7% | -- | 3 | 2.8% |
| **TOTAL** | 1.9% | 8.4% | 26.2% | 29.9% | 29.0% | 4.7% | 107 | 100.0% |

In the evaluation of the general diagnosis (Table 6), based on the time spent using digital technologies and virtual environments in teaching and learning activities, the previous pattern of the highest concentration of teachers at advanced level C1 is repeated, that is, the highest values were obtained by teachers who have been using digital for more than 14 years.

**Table 6.** Levels of digital competences of the general diagnosis by time of use of technologies and virtual environments in the teaching and learning process.

| Time of Use | DIGITAL SKILLS LEVEL (GENERAL DIAGNOSIS) | | | | | | TOTAL | |
| | A1: Newcomer | A2: Explorer | B1: Integrator | B2: Specialist | C1: Leader | C2: Pioneer | N | % |
|---|---|---|---|---|---|---|---|---|
| 1 to 3 years | 16.7% | 33.3% | 16.7% | 33.3% | -- | -- | 6 | 5.6% |
| 4 to 6 years | -- | -- | 40.0% | 40.0% | 20.0% | -- | 5 | 4.7% |
| 7 to 10 years | -- | 16.7% | 22.2% | 38.9% | 22.2% | -- | 18 | 16.8% |
| 11 to 13 years | 5.0% | -- | 55.0% | 25.0% | 15.0% | -- | 20 | 18.7% |
| 14 to 16 years | -- | 5.9% | 11.8% | 35.3% | 41.2% | 5.9% | 17 | 15.9% |
| Over 16 years | -- | 7.3% | 19.5% | 24.4% | 39.0% | 9.8% | 41 | 38.3% |
| **TOTAL** | 1.9% | 8.4% | 26.2% | 29.9% | 29.0% | 4.7% | 107 | 100.0% |

As might be expected, teachers with fewer years of incorporating digital into their practices obtained lower results, being placed at levels A and B, with none at level C, and only teachers who have been using digital for more than 14 years being placed at level C2. The analysis also shows that the distribution of the general diagnostic score reveals statistically significant differences between time of use of technologies ($p < 0.001$), 1 to 3 years versus more than 16 years, and between 11 and 13 years versus more than 16 years, with the group of more than 6 years presenting higher scores (Figure 10).

Regarding the distribution by areas, the score for Area 4 (Assessment) shows that there are statistically significant differences between time of use of technologies ($p = 0.003$), 11 to 13 years versus 14 to 16 years, and between 7 and 10 years versus 14 to 16 years, with higher times of use having higher scores (Figure 11).

In Area 5, the distribution of the score (Empowering Learners) reveals that there are statistically significant differences between time of use of technologies ($p = 0.004$), 1 to 3 years versus more than 14 years, and between 11 and 13 years versus more than 16 years, with higher times of use having higher scores (Figure 12).

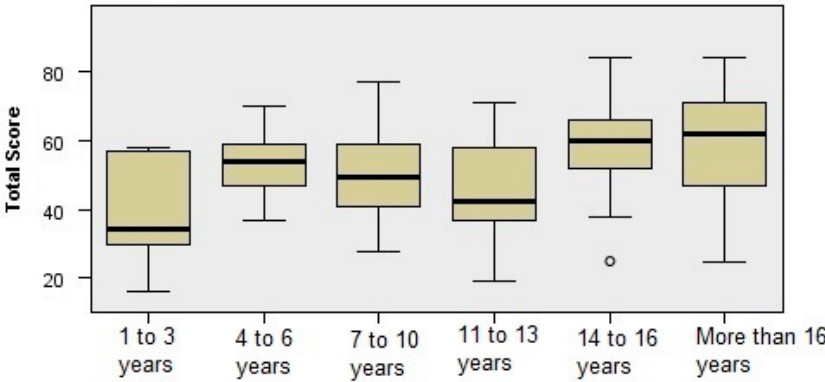

**Figure 10.** Distribution of the score of the general diagnosis by time spent using digital technologies and virtual environments (Kruskal–Wallis test, $p < 0.001$).

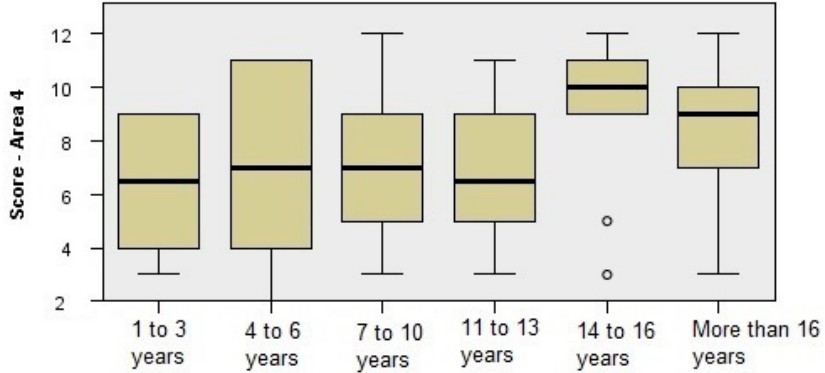

**Figure 11.** Distribution of the score of Area 4 (Assessment) by time spent using digital technologies and virtual environments (Kruskal–Wallis test, $p = 0.003$).

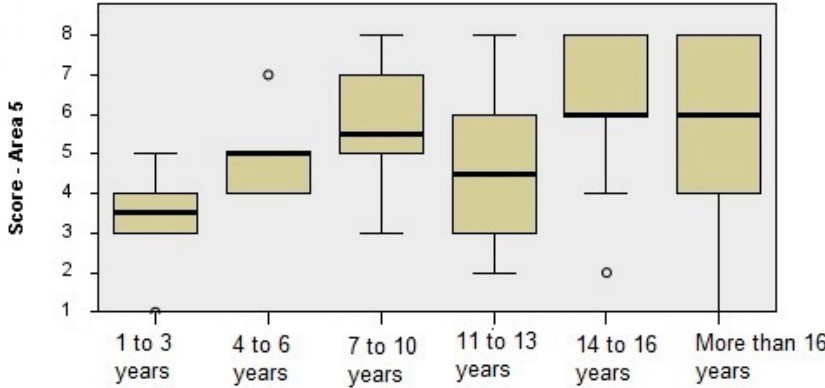

**Figure 12.** Distribution of the score of Area 5 (Student Empowerment) by time spent using digital technologies and virtual environments (Kruskal–Wallis test, $p = 0.004$).

Finally, the distribution of the score for Area 6 (Promoting Students' Digital Competence) reveals that there are statistically significant differences between time of use of technologies ($p = 0.047$), 11 to 13 years versus more than 16 years, that the latter of which presents a higher score (Figure 13).

When describing the general diagnosis of digital competences by organic unit (Table 7), the pattern is repeated once again, with the highest concentration of teachers at the advanced level C1 (Leader) being located in the departments of Science and Technology (36.1%) and Education and Distance Learning (36%), i.e., departments with teachers whose initial or postgraduate training and/or research has a direct relationship with the areas

of the digital and education, namely at the level of distance education. In turn, the departments of Social Sciences and Management and Humanities also present near-identical results, with 39%, 1% and 36%, respectively, but with the highest values located one level below, at level B2 (Specialist).

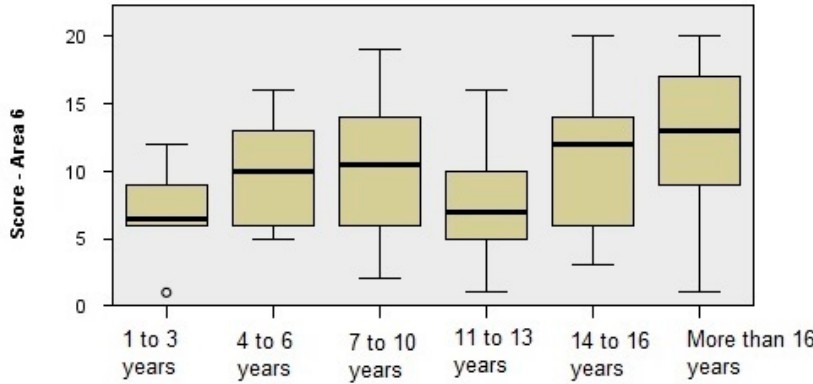

**Figure 13.** Distribution of the score of Area 6 (Promoting Students' Digital Competence) by time spent using digital technologies and virtual environments (Kruskal–Wallis test, $p = 0.047$).

**Table 7.** Digital competence levels of the general diagnosis by organic unit.

| Organic Unit | DIGITAL SKILLS LEVEL (GENERAL DIAGNOSIS) | | | | | | TOTAL | |
| --- | --- | --- | --- | --- | --- | --- | --- | --- |
| | A1: Newcomer | A2: Explorer | B1: Integrator | B2: Specialist | C1: Leader | C2: Pioneer | N | % |
| Department of Science and Technology (DCeT) | -- | 2.8% | 25.0% | 27.8% | 36.1% | 8.3% | 36 | 33.6% |
| Department of Social and Management Sciences (DCSG) | 4.3% | 13.0% | 26.1% | 39.1% | 17.4% | -- | 23 | 21.5% |
| Department of Education and Distance Learning (DEED) | 4.0% | 8.0% | 24.0% | 20.0% | 36.0% | 8.0% | 25 | 23.4% |
| Department of Humanities (DH) | -- | 13.6% | 31.8% | 36.4% | 18.2% | -- | 22 | 20.6% |
| Lifelong Learning Unit (UALV) | -- | -- | -- | -- | 100.0% | -- | 1 | 0.9% |
| **TOTAL** | 1.9% | 8.4% | 26.2% | 29.9% | 29.0% | 4.7% | 107 | 100.0% |

In the inferential analysis of the results related to the Organic Unit, the only respondent from the Lifelong Learning Unit (UALV) was removed, considering only 106 responses.

This analysis also highlights that the distribution of the general diagnosis score and reveals statistically significant differences between the organic units ($p = 0.035$), namely between DCeT and DCSG, with DCeT presenting a higher score (Figure 14).

Regarding the distribution by areas, the score of Area 1 (Professional Engagement) reveals statistically significant differences between DCeT and DCSG, and DCSG and DEED, with DCSG presenting lower scores (Figure 15).

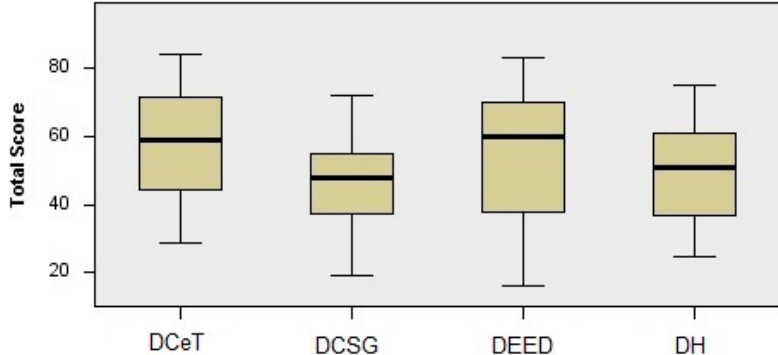

**Figure 14.** Distribution of the general diagnosis score by Organic Unit (Kruskal–Wallis test, $p = 0.035$).

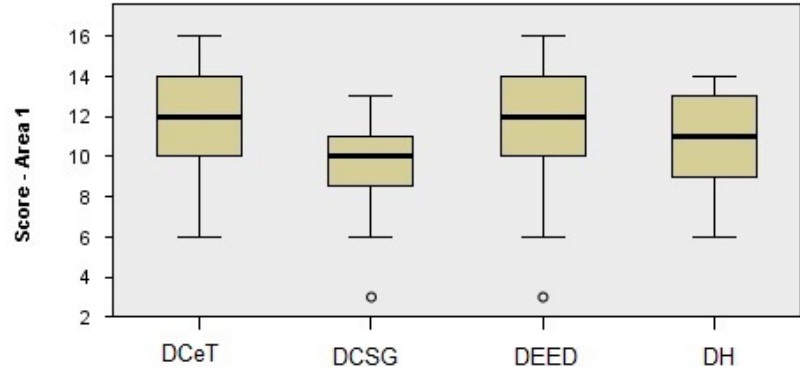

**Figure 15.** Distribution of the score of Area 1 (Professional Engagement) by Organic Unit (Kruskal–Wallis test, $p = 0.015$).

In Area 2 (Digital Technologies Resources), the score distribution shows statistically significant differences between DCeT and DCSG, with DCeT having a higher score (Figure 16).

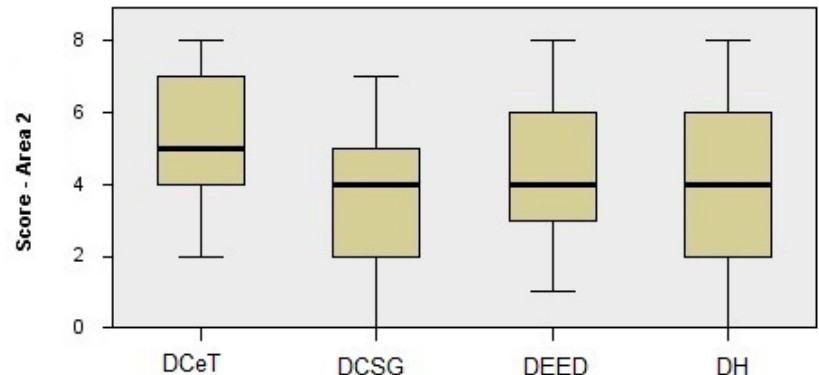

**Figure 16.** Distribution of the score of Area 2 (Digital Technologies Resources) by Organic Unit (Kruskal Wallis test, $p = 0.011$).

In turn, in Area 3 (Teaching and Learning) the score distribution reveals statistically significant differences between DCeT and DCSG, and DCSG and DEED, with DCSG showing lower scores (Figure 17).

Finally, the score distribution of Area 4 (Assessment) reveals statistically significant differences between DCeT and DCSG, and DCeT and DH, with DCeT showing higher scores (Figure 18).

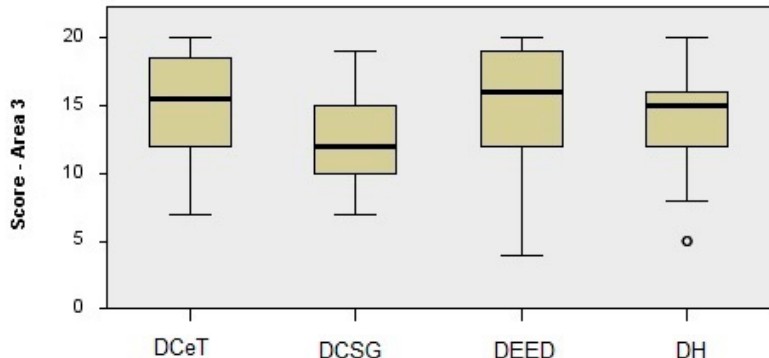

**Figure 17.** Distribution of the score of Area 3 (Learning and Teaching) by Organic Unit (Kruskal–Wallis test, $p = 0.038$).

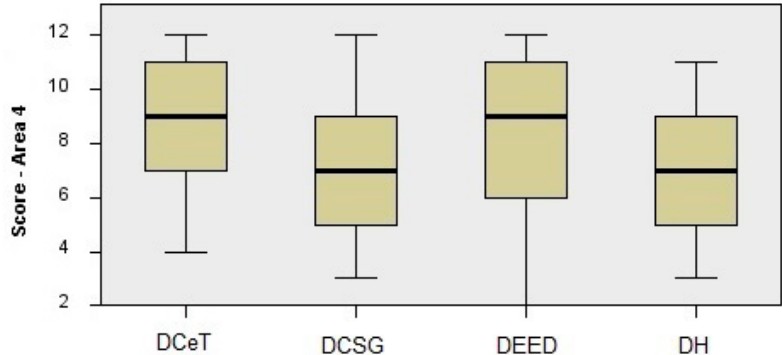

**Figure 18.** Distribution of the score of Area 4 (Assessment) by Organic Unit (Kruskal–Wallis test, $p = 0.042$).

### 5. Discussion and Final Considerations

The main finding of this study indicates that UAb teachers possess a high level of proficiency in digital competence, as per the B2 Expert level outlined in DigCompEdu. The average proficiency lies between the B2 Expert and C1 Leader levels, demonstrating a confident, creative, and critical integration of digital tools in pedagogical practices. However, despite these positive results, two of the core areas of DigCompEdu, namely Competence Areas 2—Digital Resources, and 4—Assessment, exhibited relatively lower values and demand special attention. Similarly, Competence Area 6—Promoting Students' Digital Competence, also displayed room for improvement.

To address this situation, it becomes crucial to focus on further enhancing proficiency levels through targeted training efforts. In Competence Area 2, specific actions should be developed to cultivate appropriate search strategies for identifying and selecting quality digital resources, aligning them with the context and learning objectives. Teachers should also learn to critically evaluate the credibility and reliability of digital sources, while being mindful of any usage restrictions like copyright, file type, technical requirements, legal provisions, and accessibility.

Regarding Area 4—Assessment, it is vital to promote capacity-building actions that analyze diverse assessment formats and approaches (diagnostic, formative, and summative). Moreover, critical evaluation of the evidence gathered through digital technologies on students' activity, performance and progress should be emphasized.

Moving on to Area 6—Promoting students' digital competence, which exhibited the lowest results among all areas, capacity-building actions should concentrate on planning and implementing tasks that encourage students to use digital technologies for critical information evaluation and management. Additionally, fostering respectful communication and collaboration in digital environments, along with enabling students to create digital content while respecting copyright rules, are key objectives.

Since this study was requested by the Rectorate of UAb as part of a policy of active training of its teachers in the area of online teaching, we will therefore present these results and define a training plan that allows the development of digital competences in a more consistent way in the areas that presented lower values. By investing in these training actions, the university can continue to foster innovation and advancement in digital competence among its educators.

The study also highlights heterogeneity in proficiency levels among the university's departments. Science and Technology, as well as Education and Distance Learning, scored at level C1 (Leader) with very similar values of 36.1% and 36%, respectively. Social Sciences and Management and Humanities departments achieved slightly higher results, 39.1% and 36%, respectively, at level B2 (Expert). This disparity can be attributed to the inherent nature of Distance Education, where UAb teachers regularly interact with virtual platforms and digital technologies. The higher proficiency of the first two departments may be due to their intrinsic association with digital and distance education, both in research and initial or postgraduate training levels.

Comparatively, this study found that the proficiency levels of Higher Education teachers at UAb are higher, in contrast to other studies conducted in Portugal, where teachers were at level B1—Integrator, two levels below the results here. The studies developed with Higher Education teachers in Portugal [6,11,12] reveal this significant difference, since in these studies the teachers are at level B1—Integrator, practically two levels below the results presented in the present study with teachers from UAb. It is interesting to note that the study by Santos, Pedro and Mattar [11] that seeks to draw a national portrait of the digital competences of Higher Education teachers in Portugal reveals that teachers who teach in the Distance Education modality (n = 42), certainly the majority of teachers from the UAb, are at level B2—Specialist.

It is also interesting to note that, unlike other studies previously presented, the results of the present research contradict this trend, since teachers in the older age groups are those with higher proficiency levels, and studies carried out in Spain and other countries show similar results to those in Portugal [13–16]. The age group of older teachers (60 or more years old) is the one with the highest value at level C2 (Pioneer), and the highest concentration of teachers is found at levels B2 (Expert) and C1 (Leader), with emphasis on the 40 to 49 age group with a value of 43.8% at level B2, and the 50 to 59 age group with a value of 36.7%. These results are aligned with the variable length of service, since at this level it is also concluded that teachers with more time in service (more than 36 years) are those who present higher percentage values, being located at the advanced level C1. Since the teaching and learning activities of UAb teachers are developed on digital platforms, it would be expected that teachers with more time in service and more years of using digital technologies would have the highest scores. Indeed, the highest scores on this question were obtained by teachers who have been using digital for more than 14 years, with the highest concentration of responses placing teachers at level C1. On the other hand, and as would be expected, teachers with fewer years of incorporating the digital into their practices obtained lower results, being exclusively at levels A and B. In reality, these results are not surprising, because the natural "habitat" of these teachers who have been working since the beginning of the 21st century in the UAb are digital and virtual environments.

In conclusion, it is essential to consider certain aspects not previously mentioned when interpreting these results. Firstly, the questionnaire's typology relies on teachers' perceptions rather than practical demonstrations of knowledge, and its correlation with the variables discussed earlier. Secondly, although the results are presented in an aggregated format, they are available individually for each teacher (via the individual feedback report) and for each department within the institution. Consequently, this study provides insights at both the individual and institutional levels.

At the individual level, the personalized feedback report reveals each teacher's specific digital competences, highlighting strengths and areas for improvement in particular

competence areas or skills. This empowers teachers to reflect on their practices and plan their professional development accordingly.

At the institutional level, the results offer valuable data for the university's leadership bodies to assess their teachers' digital competences. This information informs decisions regarding priority areas for professional development and helps create tailored plans. Encouraging collaboration among teachers is also crucial, with those at higher proficiency levels (Leader and Pioneer) supporting those with lower proficiency levels. By monitoring teachers' competences and practices over time, the institution can establish targets and assess the effectiveness of their professional development initiatives. These practices foster a continuous improvement cycle within the university's digital education landscape.

This study carries out only a quantitative approach, which may be a limitation. An approach that also considered a qualitative study, with interviews with a small group of teachers, could have brought new insights into the digital competences of these teachers.

As a suggestion for future work, we think that, after the development of the training courses in the areas that were considered most fragile, it would be important to administer the questionnaire again to assess the impact of these courses.

Informed consent was obtained from the participants before the study began. The participants were assured that their participation was voluntary and that they could withdraw from the study at any time, and that the data collected from the participants was kept confidential and anonymous and would only be used for research purposes. This research project is in line with the Ethical Charter published by the Portuguese Society of Education Sciences and follows the guidelines linked to it.

**Author Contributions:** All authors contributed to the design and implementation of the research, to the analysis of the results and to the writing of the manuscript. All authors have read and agreed to the published version of the manuscript.

**Funding:** This research received no external funding.

**Data Availability Statement:** Not applicable.

**Conflicts of Interest:** The authors declare no conflict of interest.

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
