# Peer review of "Digital Competence of Higher Education Teachers at a Distance Learning University in Portugal"

_computers, doi:10.3390/computers12090169_

Round 1
Reviewer 1 Report
The paper presents well-designed research wich results that may be of interest to readers. However, I believe that some modifications in the presentation of the results are necessary.
I present them below and I hope they will be useful for the authors.
1. Reorganization of the text.
There are some fragments of the text that should be relocated to sections more appropriate to your content.
For example, lines 93-96 are more appropriate for the "methods" section than the introduction.
Similarly, lines 293-295 could be included in the "3.1 Questionnaire" section.
Lines 481-482 are more appropriate from the discussion section.
2. Image quality of the figures.
Figures 1 and 2 (especially 2) have low quality. It is suggested to replace them with sharper images.
3. Table 1.
The values ​​in Table 1 are not those used in the research presented in the paper. It is suggested to delete the table and briefly summarize its information in text format. Leaving the table as it is now causes some confusion with the content of table 2.
4. Figures 3 and 4.
I think these figures are unnecessary. These data can simply be provided in text format.
5. Figures 5 and 6.
They also seem unnecessary. These data could be provided in a single table.
In general, the number of figures is too high for what is usual in a paper of this type.
6. Table 3.
The Cumulative percentage column seems unnecessary. My recommendation is to remove it.
In this table, it would be interesting to include the number of professors in each department. In other words, the number of teachers from each department that completed the questionnaire is now indicated. But we don't know if this number is proportional to the total number of professors in each department.
In the event that there is a department in which the response rate is significantly high or low (compared to the other departments), this data should be known, as it may affect the interpretation of the results.
7. Tables 4-7.
It must be specified if there is a statistical reason to mark some values ​​with their shaded boxes.
8. Figure 14.
I think this figure does not provide great value. It is enough with what is indicated in table 5.
9. Kruskal-Wallis test.
It indicates that this statistic is calculated. But no results are provided.
The results should be added, as they can help to better interpret the tables and figures.
This is the issue that I think is most significant and needs to be improved.
10. Ethical considerations.
It should be indicated what ethical considerations have been followed. Similarly, if you have the endorsement of an ethics committee, it should be indicated.
11. Study limitations.
At the end of the discussion section, some of the limitations of the study should be noted (for example, only quantitative, not qualitative, methods have been used).
12. Future avenues of research.
Likewise, at the end of the discussion, some possible future lines of research derived from the results of the study could be indicated.
Reviewer 2 Report
Dear Authors,
Thank you for submitting your study for review. After a thorough examination of the manuscript, I believe that it is suitable for publication, pending minor revisions. I have provided specific comments throughout the manuscript to guide these revisions and enhance the overall clarity and quality of the paper

Reviewer 3 Report
Good article, focus on the study. I think the approach of VPM is important, but also the study of new VPM after COVID. Another important aspects will be VPM applied to labs like robotics, Electronics, for example the use of simulators.
Good article, focus on the study. I think the approach of VPM is important, but also the study of new VPM after COVID. Another important aspects will be VPM applied to labs like robotics, Electronics, for example the use of simulators. Figure 1 and 2 low quality, need to change.Some possible future lines of research need to be define.
Author Response
Dear reviewer, thank you very much for your comments and very positive assessment of the text. A number of changes have been made to the text in the light of the reviewers' comments, including suggestions for future work.
Round 2
Reviewer 1 Report
All the suggestions that I raised in my first report have been correctly addressed.
I think the authors have done a good job.